# Evaluation of River Water Quality Index Using Remote Sensing and Artificial Intelligence Models

Mohammad Najafzadeh * and Sajad Basirian 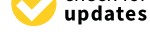

Department of Water Engineering, Faculty of Civil and Surveying Engineering,
Graduate University of Advanced Technology, Kerman 76315117, Iran
* Correspondence: m.najafzadeh@kgut.ac.ir or moha.najafzadeh@gmail.com

**Abstract:** To restrict the entry of polluting components into water bodies, particularly rivers, it is critical to undertake timely monitoring and make rapid choices. Traditional techniques of assessing water quality are typically costly and time-consuming. With the advent of remote sensing technologies and the availability of high-resolution satellite images in recent years, a significant opportunity for water quality monitoring has arisen. In this study, the water quality index (WQI) for the Hudson River has been estimated using Landsat 8 OLI-TIRS images and four Artificial Intelligence (AI) models, such as M5 Model Tree (MT), Multivariate Adaptive Regression Spline (MARS), Gene Expression Programming (GEP), and Evolutionary Polynomial Regression (EPR). In this way, 13 water quality parameters (WQPs) (i.e., Turbidity, Sulfate, Sodium, Potassium, Hardness, Fluoride, Dissolved Oxygen, Chloride, Arsenic, Alkalinity, pH, Nitrate, and Magnesium) were measured between 14 March 2021 and 16 June 2021 at a site near Poughkeepsie, New York. First, Multiple Linear Regression (MLR) models were created between these WQPs parameters and the spectral indices of Landsat 8 OLI-TIRS images, and then, the most correlated spectral indices were selected as input variables of AI models. With reference to the measured values of WQPs, the WQI was determined according to the Canadian Council of Ministers of the Environment (CCME) guidelines. After that, AI models were developed through the training and testing stages, and then estimated values of WQI were compared to the actual values. The results of the AI models' performance showed that the MARS model had the best performance among the other AI models for monitoring WQI. The results demonstrated the high effectiveness and power of estimating WQI utilizing a combination of satellite images and artificial intelligence models.

**Keywords:** water quality index; remote sensing; artificial intelligence models; spectral bands; spectral indices; natural streams

## 1. Introduction

At the present time, human access to sanitary water resources has become cornerstone of great importance for various consumptions. With the increase in urbanization along the rivers and uneven distribution of water treatment centers, many areas have been facing barriers to reducing surface water quality. Monitoring water resources plays a crucial role in human health and preserves the ecosystem, and consequently, reducing the water quality causes irreparable effects on humans and the environment. Hence, it is necessary to continuously monitor the water quality states of natural streams [1–4]. Water Quality Index (WQI) is a straightforward and mathematical equation that has been employed by Horton [5] and Brown [6] to demonstrate suitable water quality conditions for agricultural, industrial, and drinking purposes. In order to compute values of WQI, a variety of Water Quality Parameters (WQPs) need to be observed/measured [3–6]. Accessing the measured data of water quality parameters is one of the attention-stricken obstacles that environmentalists face. Although traditional methodologies of monitoring water quality are based on in situ measurement (for a point in time and space), which provides accurate observations,

these are laborious, expensive, and time-consuming [7–9]. The recent advancement in remote sensing technologies brings vast opportunities to identify and quantify WQPs [9]. Over the last decade, the capacity of Landsat-8 [10–12] and Sentinel 2 [10,11,13,14] has been evaluated to identify states of WQPs in several studies.

In recent years, numerous scholars have carried out investigations to create an empirical algorithm to estimate WQP (optically active) using satellite images, such as Turbidity [10,15,16] and chlorophyll-a [16–18].

### 1.1. Literature Review

Moreover, with the state-of-the-art development in machine learning (ML), data mining (DM), and deep learning (DL) techniques, applying AI techniques to perform various analyses of satellite images have improved the monitoring of water bodies' quality in recent years [19–21]. Numerous studies have been done to estimate various WQPs with the aid of satellite images and AI techniques.

Chebud et al. [22] employed Landsat TM images and ground-measured data to develop the artificial neural network (ANN) model in order to estimate three water quality parameters (chlorophyll-a, turbidity, and phosphorus) in the Kissimmee River basin located in the state of Florida, USA. According to the results of their research, the developed model had a high ability to monitor water quality parameters. In another study, genetic programming (GP) was successfully used by Chang et al. [23] to create a connection between the total phosphorus (TP) concentration data and the moderate-resolution imaging spectroradiometer (MODIS) images for Tampa Bay located in the state of Florida, USA. Kim et al. [24] used geostationary ocean color imager (GOCI) satellite images to monitor Chl-a and suspended particulate matter (SPM) concentrations on the west coast of South Korea. For this purpose, three machine learning models of random forest (RF), cubist regression model (CRM), and support vector machine (SVR) were used to understand the robust correlation between the measured values of WQPs and GOCI image data. The results showed that the SVR model was superior over other ML models. Later, Sharaf El Din et al. [25] indicated the successful performance of a back propagation neural network (BPNN) to predict turbidity, total suspended solid (TSS), chemical oxygen demand (COD), biochemical oxygen demand (BOD), and dissolved oxygen (DO). The monitored WQPs had been collected from Saint John River, Canada and additionally, satellite images of Landsat-8 were used. From their study, $R^2$ (coefficient of determination) values were 0.991, 9.933, 0.937, 0.93, and 0.934 for turbidity, TSS, COD, BOD, and DO, respectively. Arias-Rodriguez et al. [26] used extreme learning machine (ELM), support vector regression (SVR), and linear regression (LR) models in order to estimate Chl-a, turbidity, total suspended matter (TSM), and Secchi disk depth (SDD) using the national water quality monitoring system. Through their research, they applied data from Mexico for four lakes and additionally, Landsat-8 OLI, Sentinel-3 OLCI, and Sentinel-2 MSI were recruited. They found that the ELM model had relatively better performance in water quality estimation than other machine learning models.

Moreover, Najafzadeh et al. [3] predicted monthly WQI by 12 WQPs, such as $Ca^{2+}$, $Na^+$, $Mg^{2+}$, pH, COD, BOD, DO, electrical conductivity (EC), total hardness (TH), phosphate ($PO_4^{3-}$), nitrate ($NO_3^-$), fecal coliform (FC), turbidity (Tur), and ammonium ($NH_4^+$) for the Karun River, Iran. They obtained monthly temperature values from satellite images of Landsat-7. From their research, it was found that the MT model had the best performance for the estimate of water quality index (WQI) compared with multivariate adaptive regression spline (MARS), gene expression programming (GEP), and evolutionary polynomial regression (EPR).

In Hassan et al.'s [7] research, the structure of an artificial neural network (ANN) was optimized by a bio-inspired technique called a binary whale optimization algorithm (BWOA) to determine the relationship between the satellite reflection value Sentinel-2 and the observed values in order to estimate optically active and non-optically active parameters. They used the field data from Nasser Lake, Egypt, and the Bin El Ouidane Reservoir, Morocco. BWOA-ANN models obtained coefficients of determination ($R^2$)

of 0.916 and 0.890 for both an estimation of optically active and non-optically active parameters, respectively. On the other hand, Hong et al. [19] employed four improved structures of deep neural network (DNN) models (ResNet-18, ResNet-101, GoogLeNet, and Inception v3) to establish a relationship between hyperspectral imagery captured by drones and in situ measurements, as well as meteoroidal data to monitor Chl-a, phycocyanin (PC), and turbidity for Daechung Dam reservoir, South Korea. The results of their study demonstrated that the ResNet-18 model had the most accurate performance out of the other DNN models.

Ahmed et al. [20] employed four types of ANN models (i.e., convolutional neural network [CNN], fully connected network [FCN], multi-layer perceptron [MLP], and recurrent neural network [RNN]) and six structures of long short-term memory (LSTM) model (i.e., LSTM-Dominated, Vanilla-LSTM, Stacked-LSTM, Bidirectional-LSTM, Convolutional-LSTM, and CNN-LSTM) in order to monitor concentrations of DO and electrical conductivity (EC) parameters using shuttle radar topography mission (SRTM) data for the Rawal watershed stream network, Pakistan. They found that the bidirectional-LSTM has better performance in predicting DO and EC parameters. Recently, Chen et al. [27] designed a novel self-optimizing machine learning monitoring method in order to predict Chl-a, Ammonia Nitrogen (NH3-N), and Turbidity parameters. The WQPs were acquired from the Nanfei River located at Yangtze River Basin, China. Chen et al. [27] used unmanned aerial vehicle (UAV) images for analyses of WQPs. Additionally, the performance of the proposed ML model was compared with CatBoost, XGBoos, AdaBoost, random forest (RF), k-nearest neighbors (KNN), DNN, and LR models. From their research, it was found that the proposed model had the best performance.

According to the above-mentioned literature review, while there are quite a few advantages to the usability of remote sensing in this area, there are also some shortcomings that must be considered. Remote sensing techniques have four major advantages: non-invasive observations, large area coverage, high-resolution images, and real-time monitoring. In the case of shortcomings, remote sensing techniques have limitations in detecting certain water quality parameters, atmospheric interference (e.g., cloud cover, aerosols, and water vapor), limited spectral range, and expensive technologies. Overall, the use of remote sensing for water quality monitoring is a promising area of research, but there are still limitations and challenges that need to be addressed. Continued advances in technology and the development of AI techniques are likely to improve the accuracy and accessibility of remote sensing data for water quality monitoring in the future.

### 1.2. Objectives and Research Organization

The purpose of this study is to develop empirical relationships based on AI models and an analysis of remote sensing data in order to estimate WQI in the Hudson River, New York in the USA. The major contribution of this study is that applying information on spectral bands to establish conceptual multivariate regression models for each WQP and then the most influential spectral bands for approximation of WQPs is yielded. Additionally, four AI models (i.e., GEP, MT, EPR, and MARS) are developed by using spectral band properties in order to estimate WQI. This study applies AI models that are capable of providing linear and non-linear regression equations with a high degree of interpretability in comparison with previous investigations [28–31]. In fact, the majority of AI models that have been used in the literature function as a black box. Moreover, the present study does not employ optically active and non-optically active WQPs in order to monitor water quality states of natural streams in comparison with related works [7,14,20]. More importantly, this study uses properties of spectral bands to conceptually establish relationships with WQI, and as a result, this methodology would be more beneficial than studies that use optically and non-optically active WQPs.

This study is organized as follows. Section 2 describes the study area, statistical characterizations of WQPs, and computation of WQI. Section 3 is dedicated to preparing satellite data for various purposes, such as the extraction of radiation and reflectance bands

and separating water bodies from other parts of satellite images. Section 4 determines the multivariate linear correlations between Landsat 8 spectral indices and WQPs, and then, the relationships are used to estimate each WQP. After that, the key spectral bands are considered input variables to feed AI models. In Section 5, AI models are trained and tested to determine WQI. Finally, the results of AI models are statistically evaluated and compared with relevant literature.

## 2. Overview of Case Study and Water Quality Data Description

Geographically, the Hudson River rises in the Adirondack Mountains, runs through the Hudson Valley southward to the upper New York Bay between New York City and Jersey City, and then empties into the Atlantic Ocean at New York Harbor. In terms of length, the Hudson River is a 315 mile (507 km) river that flows from north to south. The watershed of the Hudson River is dominated by 13,400 square miles. The largest city where the Hudson River flows is New York, with a population size of 18.8 million, where it stands as the most populous city in the USA. Figure 1 illustrated the geographical location of the case study.

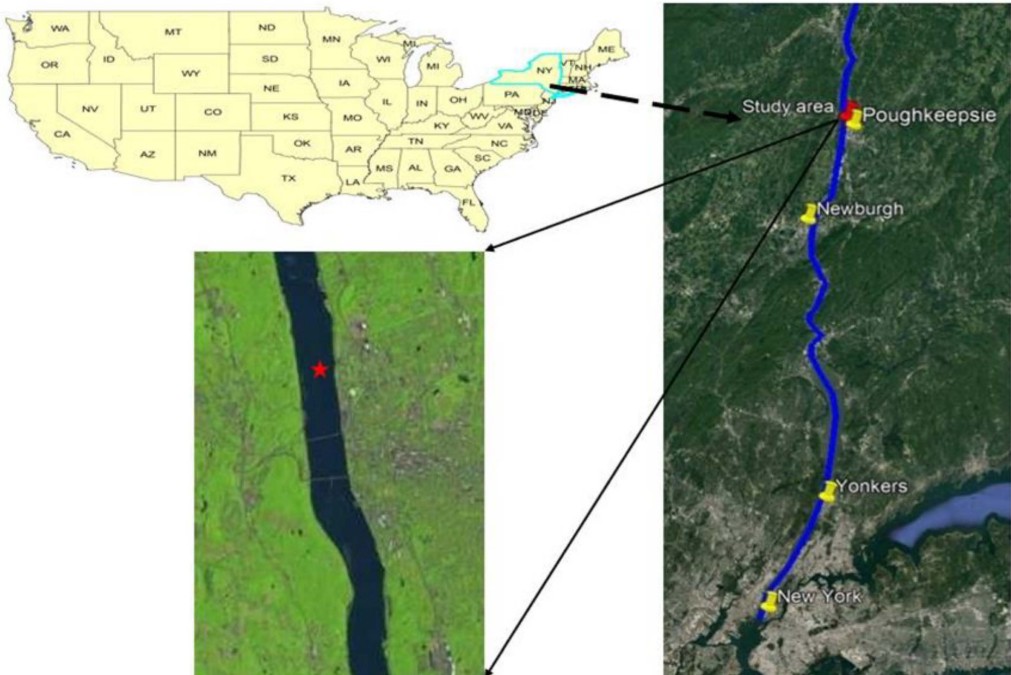

**Figure 1.** Overview of geographical localization of monitoring site for the Hudson River.

Two types of data were employed in this study: observation data and remote sensing images. The observation data includes 13 WQPs: turbidity, sulfate ($SO_4^{2-}$), sodium ($Na^+$), potassium ($K^+$), hardness, fluoride ($F^-$), dissolved oxygen (DO, chloride ($Cl^-$), arsenic (AS), alkalinity, pH, nitrate ($NO_3^-$), and magnesium ($Mg^{2+}$) whose statistical properties were listed in Table 1. These parameters were collected from the Hudson River near the Poughkeepsie, NY site located on the Hudson River with latitude 41.72176015 and longitude 73.94069299 from 14 March 2021 to 16 June 2021. The water quality data was taken from https://waterdata.usgs.gov (accessed on 11 August 2022). Additionally, some WQPs affecting sewage discharge (e.g., COD and phosphorous) were not available at the dates of the satellite images, whereas nitrate was included.

**Table 1.** Descriptive statistics of measured WQPs through the sampling point at the Hudson River.

| Parameter | Unit | Max | Min | Average | Standard Deviation |
|---|---|---|---|---|---|
| Tur | NTU | 28.69 | 1.12 | 16.67 | 6.3 |
| $SO_4^{2-}$ | mg/L | 16.7 | 9.02 | 11.66 | 2.34 |
| $Na^+$ | mg/L | 31.4 | 14.1 | 20.64 | 5.02 |
| $K^+$ | mg/L | 1.44 | 0.79 | 1.14 | 0.32 |
| pH | — | 7.9 | 7.5 | 7.57 | 0.14 |
| $NO_3^-$ | mg/L | 0.76 | 0.34 | 0.48 | 0.12 |
| $Mg^{2+}$ | mg/L | 5.76 | 3.56 | 4.64 | 0.68 |
| Hardness | mg/L | 103 | 65 | 83.1 | 10.94 |
| $F^-$ | mg/L | 0.1 | 0.1 | 0.1 | $1.9 \times 10^{-16}$ |
| $Cl^-$ | mg/L | 56.3 | 23.6 | 35.25 | 10.16 |
| AS | mg/L | $53 \times 10^{-3}$ | $27 \times 10^{-3}$ | $36 \times 10^{-3}$ | 8.12 |
| Alk | mg/L | 76.7 | 52.4 | 65.7 | 6.88 |
| DO | mg/L | 14.1 | 7.5 | 10.9 | 2.18 |

Figure 2 shows the histograms of 13 WQPs in order to better understand their frequencies. Histograms provide scholars with a summary of the changes made to data collection through visual representation. As seen in Figure 2, the frequencies of WQPs demonstrate various distributions, such as symmetrical, skewed right, skewed left, and bimodal patterns. In addition to this, half of the frequency distributions follow the bimodal pattern: Figure 2a (Tur), Figure 2d ($K^+$), Figure 2f ($NO_3^-$), Figure 2g ($Mg^{2+}$), Figure 2h (Hardness), Figure 2j ($Cl^-$), and Figure 2k (AS). Moreover, the frequencies of $SO_4^{2-}$ and $Na^+$ were illustrated in Figure 2b,c that had skewed right patterns, whereas the frequencies of the pH (Figure 2e), Alk (Figure 2l), and DO (Figure 2m) parameters followed a symmetrical pattern. Figure 2i indicates that the frequency of $F^-$ parameter has no special pattern.

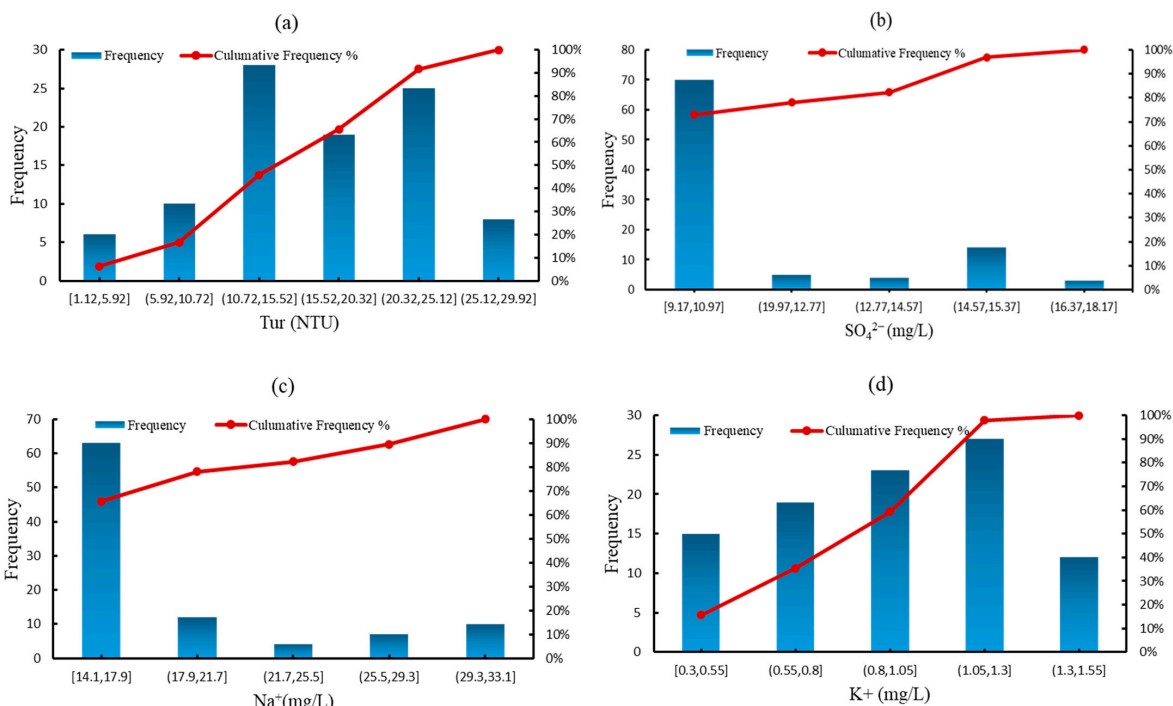

**Figure 2.** *Cont.*

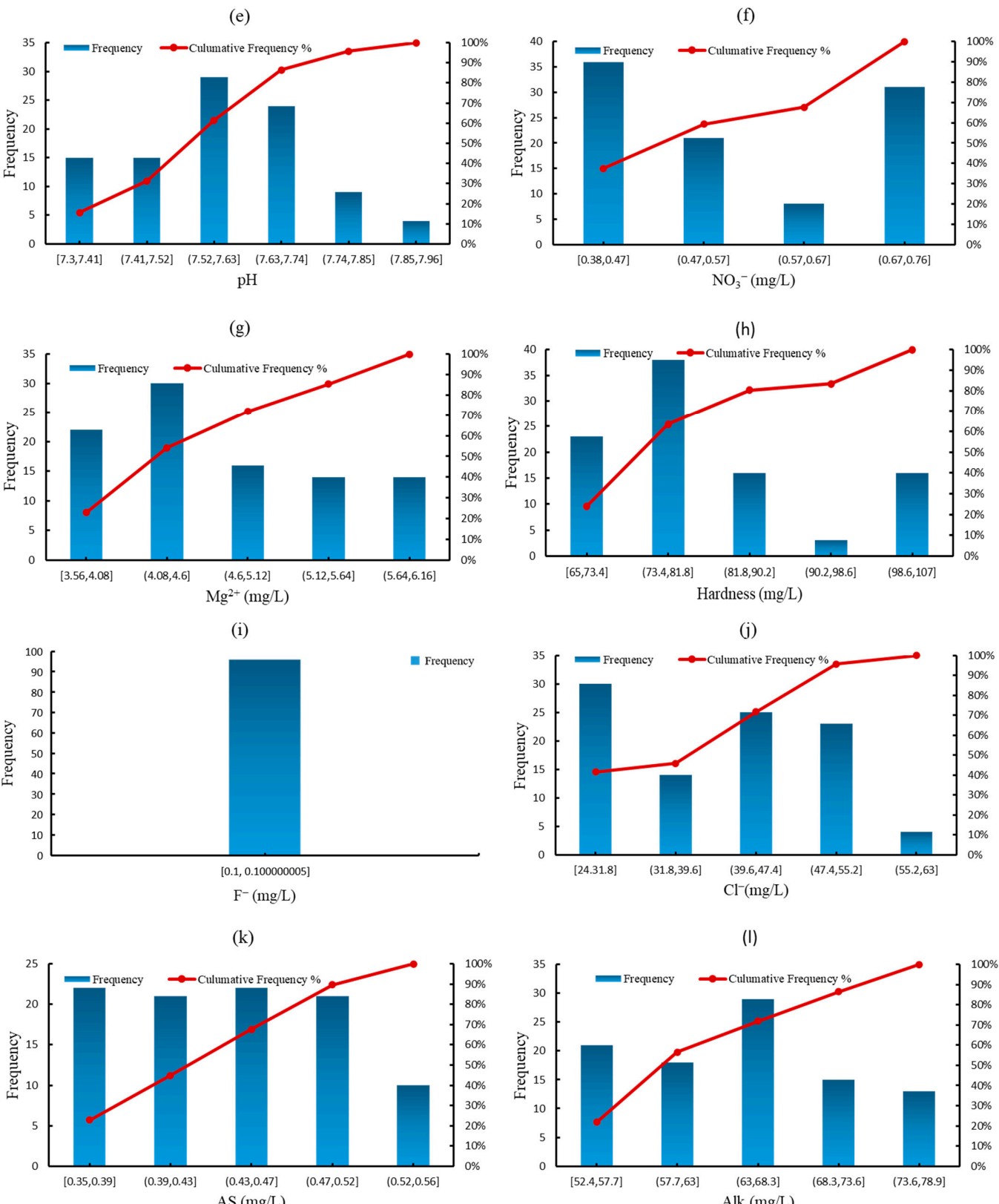

**Figure 2.** *Cont.*

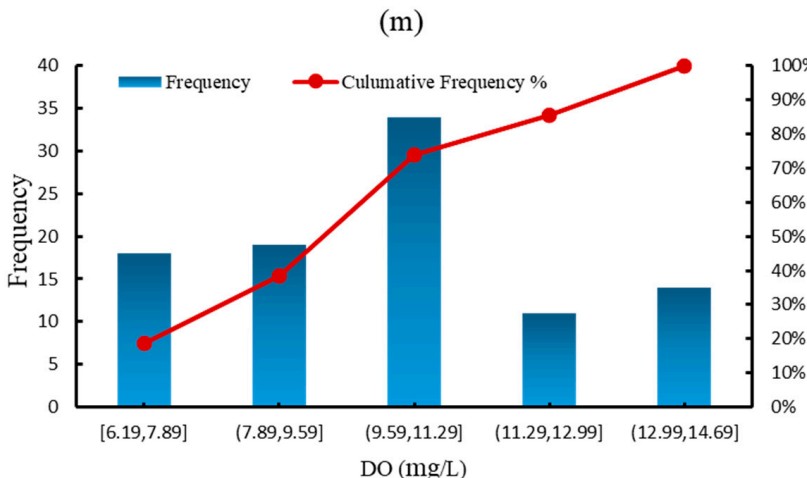

**Figure 2.** Histogram of WQPs measured in Hudson River: (**a**) tur, (**b**) $SO_4^{2-}$, (**c**) $Na^+$, (**d**) $K^+$, (**e**) pH, (**f**) $NO_3^-$, (**g**) $Mg^{2+}$, (**h**) hardness, (i) $F^-$, (**j**) $Cl^-$, (**k**) AS, (**l**) Alk, and (**m**) DO.

As seen in Table 2, 11 spectral bands with wavelengths of 0.43–11.9 μm and resolutions of 15 m (panchromatic), 30 m (visible, near-infrared [NIR], and short-wave infra-red [SWIR]), and 100 m (thermal) of Landsat-8 images have been employed. Additionally, the properties of Landsat-8 images are available at https://landsat.gsfc.nasa.gov (accessed on 11 August 2022). In this investigation, six images taken from Landsat-8 were used. These images are identical in terms of the observation date and the image capture date. Satellite images in the study area have been selected in a cloud-free state. Table 3 summarizes the properties of Landsat-8 satellite bands. The information from the previous images was utilized to fill the time gap, which ranged from 8 to 16 days between each image. Images that are taken from the USGS (United States Geographical Survey) website are available at https://earthexplorer.usgs.gov/. Other descriptions of images (i.e., identification, date, and image range) were found in Table 3.

**Table 2.** Landsat-8 Operational Land Imager (OLI) and Thermal Infrared Sensor (TIRS).

| Bands | Wavelength (μm) | Resolution (m) |
|---|---|---|
| Band 1—Coastal aerosol | 0.43–0.45 | 30 |
| Band 2—Blue | 0.45–0.51 | 30 |
| Band 3—Green | 0.53–0.59 | 30 |
| Band 4—Red | 0.64–0.67 | 30 |
| Band 5—Near Infrared (NIR) | 0.85–0.88 | 30 |
| Band 6—SWIR 1 | 1.57–1.65 | 30 |
| Band 7—SWIR 2 | 2.11–2.29 | 30 |
| Band 8—Panchromatic | 0.50–0.68 | 15 |
| Band 9—Cirrus | 1.36–1.38 | 30 |
| Band 10—Thermal Infrared (TIRS) 1 | 10.6–11.19 | 100 |
| Band 11—Thermal Infrared (TIRS) 2 | 11.50–12.51 | 100 |

**Table 3.** The details of Landsat-8 OLI-TIRS images of the Hudson River and the time frame.

| Image Acquisition Date | Image ID | Range of Image Usage |
|---|---|---|
| 12 March 2021 | LC80130312021071LGN00 | 12 March 2021 |
| 13 April 2021 | LC80140312021110LGN00 | 13 March 2021–13 April 2021 |
| 20 April 2021 | LC80140312021126LGN00 | 14 April 2021–20 April 2021 |
| 6 May 2021 | LC80130312021135LGN00 | 21 April 2021–5 May 2021 |
| 15 May 2021 | LC80140312021158LGN00 | 7 May 2021–15 May 2021 |
| 7 June 2021 | LC80130312021167LGN00 | 16 May 2021–7 June 2021 |

## 3. Data Preparation and Methods

### 3.1. Preparation of Satellite Images

In order to use satellite images, reflection from the water surface is required, and at first, the data collected in each pixel had no value or unit. Due to the fact that the images were recorded in different months, the condition of each image is different from other images, so radiometric and atmospheric correction is necessary for each of the images.

#### 3.1.1. Conversion of Digital Number to Spectral Radiance

According to Equation (1), the digital number (DN) value is converted to spectral radiation using the calibration factor of the sensor (USGS, 2016):

$$\text{L} = \text{Gain} \times \text{DN} \times \text{Offset} \tag{1}$$

where L is spectral radiance at the sensor's aperture in watts/(m$^2$ × ster × $\mu_m$), DN is the pixel value, and Gian and Offset are the sensor calibration coefficients.

#### 3.1.2. Conversion of Spectral Radiation to Spectral Reflectance

To enhance and automate ground reflectance retrieval, additional features are incorporated in this conversion:

$$\rho = \frac{\pi L d^2}{ESUN \times COS(SZ)} \tag{2}$$

where *d* is Earth–sun distance in astronomical units, *ESUN* is solar irradiance, and *SZ* is the radiation angle during satellite imaging. Correcting the atmospheric influence is a crucial next step in the image processing process. Quick atmospheric correction (QUAC) was performed using ENVI software, and the image's brightness levels were transformed to surface reflectance values.

#### 3.1.3. Separation of Water from Other Parts of Satellite Images

To investigate the relationship between satellite images and WQPs, it was necessary to separate the watershed of the Hudson River from other waterless areas. In this way, a spectral index was used [32]:

$$\text{NDWI} = \frac{\text{G} - \text{NIR}}{\text{G} + \text{NIR}} \tag{3}$$

where NDWI was the normalized difference water index, G was the green spectral band, and NIR was the near infra-red spectral band. The value of this spectral index was between +1 and −1. Parts of the image that had pure water were assigned a value of +1, and other parts without water were assigned a value between 0 and −1.

### 3.2. Correlation between Spectral Bands and WQPs

The Pearson correlation of all WQP with each of the 11 Landsat-8 bands ($b_1$, $b_2$, $b_3$, ..., $b_{11}$) was examined in SPSS software as the first step in this section of the study. The correlation coefficient (R) between the water quality parameters of the Hudson River and spectral bands is shown in Table S1 (see Supplementary Materials). The results of the Pearson correlation demonstrated that the highest correlation coefficients are listed as DO with $b_{10}$ (R= −0.914), pH with $b_{11}$ (R = 0.916), $Mg^{2+}$ with $b_{11}$ (R = 0.864), $Na^+$ with $b_9$ (R = −0.866), $SO4^{2-}$ with $b_2$ (R = −0.933), hardness with $b_1$ (R = −0.871), Alk with $b_3$ (R = 0.776), AS with $b_{10}$ (R = 0.914), $F^-$ with $b_{11}$ (R = 0.728), $K^+$ with $b_1$ (R = 0.827), $Cl^-$ with $b_{10}$ (R = −0.883), tur with $b_{11}$ (R = −0.841), and $NO_3^-$ with $b_6$ (R = −0.854).

In the next step, ratio index (RI) and normalization difference index (NDI) were calculated as:

$$NDI(R_i, R_j) = \frac{R_i - R_J}{R_i + R_J} \tag{4}$$

$$RI(R_i, R_j) = \frac{R_i}{R_J} \tag{5}$$

Random bands between 0.47 μm and 12.51 μm were chosen as $R_i$ and $R_j$ [4].

Having the spectral indices computed, the values of Pearson correlation between each spectral index and all WQPs were determined. According to related works, although using spectral indices reduces some of the noise caused by lighting and background and increases correlation coefficients with water quality parameters, the spectral index does not always increase correlation coefficients [21,33,34]. In fact, the previous investigations concluded that the correlation between ratios of spectral bands and WQPs can be considered a relatively appropriate way of approximating WQPs or WQI. Table S2 shows the correlation coefficients between water quality parameters and spectral indices (NDI and RI), as seen in the Supplementary Materials section. For instance, DO with $b_6/b_5$, pH with $b_5/b_7$, and $Mg^{2+}$ with $b_9/b_{10}$ and $b_9/b_{11}$ have the highest values of correlation coefficients.

### 3.3. Correlation between WQPs and Spectral Indices

After determining the highest correlation of single bands and spectral indices (RI and NDI), multivariate linear regression (MLR) analysis was used to create a relationship between WQPs (dependent variable) and spectral data (independent variable). Accordingly, Equation (6) expressed MLR analysis as follows:

$$WQP = A_0 + \sum_{i=1}^{k}(A_i \times X_i) \tag{6}$$

where $X_i$ was single bands and spectral indices with a high correlation coefficient and k was the number of bands; $A_0$ and $A_1$ were empirical regression coefficients obtained from in situ data observations. By applying the relationships obtained on Landsat-8 images, the value of each pixel was converted to the simulated value of WQP. This study utilizes MLR analysis in order to provide an empirical equation between WQPs and spectral indices. It was highly important to consider both single spectral indices (i.e., $b_1$, $b_2$, $b_3$, ..., $b_{11}$) and ratios of spectral indices. Table 4 presents lists of MLR equations that establish the correlation between WQPs and spectral indices. As seen in Table 4, the most correlated MLR equation (R = 0.954) was dedicated to $Cl^-$, which was approximated using $b_0$, $b_{11}$, and $b_2/b_6$, whereas the approximation of $Na^+$ has rather lower correlation (R = 0.756) with spectral indices ($b_9$, $b_{11}$, and $b_3/b_{11}$) in comparison with other WQPs. Additionally, MLR equations estimating $Cl^-$ (0.954), pH (0.939), $F^-$ (0.937), AS (0.936), and Alk (0.920) stood at the other ranks in terms of accuracy level.

**Table 4.** The most accurate equations obtained from MLR in order to estimate WQPs.

| Parameters | Multivariate Linear Regression Equation | *R* |
|---|---|---|
| Tur | $969.3 - 1.5468 \times b_{11} + 2.07 \times \frac{b_5}{b_2}$ | 0.873 |
| $SO_4^{2-}$ | $-285 + 2824 \times b_2 + 91 \times \frac{b_4}{b_3} - 548 \times b_6$ | 0.867 |
| $Na^+$ | $477 + 10,066 \times b_9 - 17.8 \times b_{11} - 34,776 \times \frac{b_3}{b_{11}}$ | 0.756 |
| $K^+$ | $1.4643 - 0.217 \times \frac{b_2}{b_6} - 0.1186 \times \frac{b_5}{b_6} + 0.0786 \times \frac{b_5}{b_7}$ | 0.849 |
| pH | $-2.03 + 0.912 \times b_{11}$ | 0.939 |
| $NO_3^-$ | $0.299 - 0.894 \times \frac{b_7 - b_5}{b_7 + b_5} - 28.17 \times b_6 - 1.31 \times \frac{b_6}{b_5}$ | 0.868 |
| $Mg^{2+}$ | $8.063 - 183,590 \times \frac{b_9}{b_{11}} + 156,519 \times \frac{b_9}{b_{11}}$ | 0.888 |
| Hardness | $-755 + 1745 + b_1 + 705 \times \frac{b_1}{b_8} + 404 \times \frac{b_4}{b_3}$ | 0.801 |
| $F^-$ | $1.597 - 0.00508 \times b_{11} - 174 \times \frac{b_9}{b_{10}}$ | 0.937 |
| $Cl^-$ | $277 + 0.0001 \times b_{10} - 0.807 \times b_{11} - 3.835 \times \frac{b_2}{b_6}$ | 0.954 |
| AS | $0.38939 - 0.02497 \times \frac{b_6}{b_5} + 0.04198 \times \frac{b_2}{b_6}$ | 0.936 |
| Alk | $-18,804 + 27,173 \times \frac{b_{11} - b_2}{b_{11} + b_2} - 8290 \times \frac{b_{11} - b_1}{b_{11} + b_1} - 0.0009 \times \frac{b_{11}}{b4}$ | 0.920 |
| DO | $103.16 - 0.3289 \times b_{11}$ | 0.917 |

*3.4. WQI Calculation*

WQI values are generally estimated by two globally accepted guidelines: NSF (National Sanitation Foundation) and CCME (Canadian Council of Ministers of the Environment) [3,35]. In this study, the WQI values given by CCME guideline were applied to approximate the water quality of the Hudson River. This method has no parameter restrictions; the more parameters that were used as inputs to this index, the more accurately the state of the water quality was assessed. The WQI values by CCME produced a dimensionless number in the range of 0 to 100, where 0 and 100 denoted the poorest and excellent quality (Canadian Council of Ministers of Environment, 2001). The maximum and minimum values of the water quality index (WQI) for the Hudson River were 96.25 and 84.2, respectively. WQI scale is shown in Table 5.

**Table 5.** Various states of WQI values based on CCME guidelines.

| Class | Threshold Value | Water Quality States |
|---|---|---|
| I | 95–100 | Excellent |
| II | 80–94 | Good |
| III | 60–79 | Fair |
| IV | 45–59 | Marginal |
| V | 0–44 | Poor |

In this study, we tried to predict WQI values based on spectral indices, and additionally, WQI was generally dependent on WQPs. After that, it was proved that there was a strong correlation between WQPs and spectral indices through MLR analysis. Therefore, it can be inferred that WQI is inextricably bound up with spectral indices. Moreover, WQI was expressed as follows,

$$WQI = f(b_1, b_2, b_3, b_4, b_5, b_6, b_7, \ b_8, b_9, b_{10}, b_{11}) \tag{7}$$

In Equation (7), statistical descriptions of input–output variables have been given in Table 6. The present study utilized single spectral bands in order to feed AI models due to

the fact that these AI models would provide the best non-linear combinations of spectral bands (i.e., regression-based equations).

**Table 6.** Descriptive statistics of single bands and WQI.

| Parameter | Max | Min | Average | Standard Deviation |
|:---:|:---:|:---:|:---:|:---:|
| $b_1$ | 0.107 | 0.034 | 0.055 | 0.02 |
| $b_2$ | 0.09 | 0.037 | 0.055 | 0.02 |
| $b_3$ | 0.072 | 0.029 | 0.048 | 0.012 |
| $b_4$ | 0.09 | 0.029 | 0.057 | 0.021 |
| $b_5$ | 0.052 | 0.025 | 0.034 | 0.007 |
| $b_6$ | 0.038 | 0.016 | 0.027 | 0.007 |
| $b_7$ | 0.053 | 0.011 | 0.026 | 0.011 |
| $b_8$ | 0.171 | 0.036 | 0.064 | 0.012 |
| $b_9$ | 0.004 | 0.0009 | 0.002 | 0.001 |
| $b_{10}$ | 293.7 | 276.22 | 283.34 | 6.13 |
| $b_{11}$ | 292.9 | 275.71 | 282.67 | 5.99 |
| WQI | 96.25 | 84.25 | 88.11 | 3.68 |

All AI models were performed using 11 input variables whose 71 dataseries (75% of dataseries) were applied to carry out the training phase, and then, the remaining dataseries (24 series of spectral indices) were allocated to perform the testing phase.

### 3.5. Definition of Statistical Indices

To investigate the evaluation of AI models efficiency in the training and testing phases, index of agreement (IOA), root mean square error (RMSE), mean absolute error (MAE), and scatter index (SI) have been utilized. These statistical criteria were frequently applied to evaluate WQI predictions and other water resources problems, such as stream flow forecasting and soil temperature (e.g., [3,25,30,36–39]).

$$IOA = 1 - \frac{\sum_{i=1}^{U}(WQI(i)_{Obs} - WQI(i)_{Pre})^2}{\sum_{i=1}^{U}\left(\left(WQI(i)_{Pre} - \overline{WQI_{Obs}}\right) - \left(WQI(i)_{Obs} - \overline{WQI_{Obs}}\right)\right)^2} \tag{8}$$

$$RMSE = \left[\frac{\sum_{i=1}^{U}(WQI(i)_{Pre} - WQI(i)_{Obs})^2}{U}\right]^{1/2} \tag{9}$$

$$MAE = \frac{1}{U}\sum_{i=1}^{U}|WQI(i)_{Pre} - WQI(i)_{Obs}| \tag{10}$$

$$SI = \frac{\sqrt{(1/U)\sum_{i=1}^{U}\left(\left(WQI(i)_{Pre} - \overline{WQI_{Pre}}\right) - \left(WQI(i)_{Obs} - \overline{WQI_{Obs}}\right)\right)^2}}{(1/U)\sum_{i=1}^{U}WQI(i)_{Obs}} \tag{11}$$

where $WQI_{Pre}$ denotes predicted values of WQI by AI models, $WQI_{Obs}$ denotes the computed values of WQI by CCME guideline, $\overline{WQI}$ is the average value of WQI, and U is the number of WQI samples.

IOA criterion was developed by Willmott [36] as a standardized measure of the degree of numerical model estimation error and varied between 0 and 1. The best value of the *IOA* value was +1. This meant that the AI model showed the best performance. Additionally, the worst value was zero, which indicated the worst performance of the test model. *RMSE*, MAE, and SI were the error function values, ranging from 0 to +∞.

## 4. Implementation of Soft Computing Models

### 4.1. Model Tree

The M5 tree model is one of the data mining methods that has received much attention in modeling various problems [3]. This model is an extension of the MT tree model proposed by Quinlan [40]. Compared to regression trees, the advantages of the M5 tree model are that it is smaller and reduces computational costs. The M5 method divides a complex problem into many subdomains, and a multivariate regression model is considered for each subdomain. Overall, MT implementation includes three steps: tree structure creation, pruning, and smoothing stages. In the first stage, a regression tree is generated based on the decision tree in which intra-subdivision variations in the class values of each branch is minimized to split the tree structure into branches, leaves, and nodes. In this way, the concept of purity plays a key role in controlling the splitting criterion. The standard deviation of each class value of data points at a certain node is considered. After the tree structure is grown, a multivariate linear regression is fitted on the data sets of class. Additionally, the pruning stage is performed to estimate "true error" for each subtree (or multivariate linear regression at each node of tree). In the smoothing stage, sharp discontinuities among adjacent multivariate linear regression models at the leave of the pruned tree are efficiently controlled. This process combines the leaf model estimation with the aid of every node (where splitting parameters are available) along the path back to the root [3,40].

The implementation of MT was conducted by Weka3.9 software. To monitor the water quality status of water bodies, the surveying on the related works proved that applying multivariate linear equations provided the best performance for training and testing phases (e.g., [3,41–43]). In this way, we used the following expression to approximate *WQI*:

$$WQI = bias + \sum_{i=1}^{11} a_i b_i \tag{12}$$

in which $a_1$, $a_2$, $a_3$,..., $a_{11}$ were a set of weighing coefficients related to Equation (12). Performance of MT indicated that 6 input variables were applied to feed M5MT. In this way, two multivariate linear equations were obtained as follows:

If $b_6 \leq 0.021$,

$$WQI = 61.7657 - 13.466\, b_2 + 14.7722\, b_4 - 30.7064\, b_6 - 35.6578\, b_7 - 215.1696\, b_9 + 0.1109\, b_{11} \tag{13}$$

Otherwise,

$$WQI = 65.4074 - 9.5862\, b_2 + 10.5158\, b_4 - 21.8588\, b_6 - 25.3835\, b_7 - 153.1716\, b_9 + 0.079\, b_{11} \tag{14}$$

In Equations (13) and (14), $b_6$ was the splitting variable, and the corresponding value was 0.021. Moreover, Equations (13) and (14) were provided using smoothing and pruning the trees.

### 4.2. Multivariate Adaptive Regression Spline

MARS is a linear regression analysis method that Friedman first proposed for solving high-dimensional problems. The MARS model is a non-parametric technique that can create polynomial expressions between the independent and response variables to analyze complex systems [44]. This model is created in two stages. In the forward step, the basic functions are entered into the model, and the nodes are selected to improve the performance, and an Overfitted model is obtained. In the backward stage, the terms that had a minor effect were eliminated one after the other based on the generalized cross-validation (GCV) value until the best model was created [3,44,45]. The MARS technique was obtained from an aggregate of basis functions (BFs):

$$WQI(a\ set\ of\ spectral\ indices) = C_0 + \sum_{i=1}^{N} WC_i.BF_i[a\ set\ of\ spectral\ indices] \tag{15}$$

in which WC$_i$, C$_0$, and N were the weighting coefficients (WCs) computed with the least squares (LS) technique, the constant coefficient (or bias), and the number of basis functions, respectively.

The MARS technique, as a programming computer-aided-simulation, was implemented using MATLAB 2008a software. To reduce the complexity of the initial adaptive regression model, the analysis of GCV was performed during the forward and backward development phases. To perform cross-validation, k-fold was equal to 10. During each fold, forward and backward stages were carried out, and then, the number of BFs and total effective parameters were yielded. Having 10 folds performed, the average of prediction results was computed. From the final step of MARS development, eleven BFs that formed regression spline equations were obtained:

$$
\begin{aligned}
WQI = {} & 85.592 + 1.1585 \times BF_1 + 876.48 \times BF_2 - 21366 \times BF_3 + 4.6102 \times 10^7 \times BF_4 + 2.2885 \times 10^6 \times BF_5 \\
& -1731.8 \times BF_6 + 1.17312 \times 10^6 \times BF_7 - 0.26142 \times BF_8 - 432.97 \times BF_9 - 1.8534 \times BF_{10} \\
& +8.1124 \times 10^5 \times BF_{11}
\end{aligned}
\tag{16}
$$

in which the regression equation consisted of BFs. These BFs were generally quadratic polynomial expressions, as seen in Table 7. Through the development of the MARS model, five spectral indices (i.e., $b_1$, $b_2$, $b_4$, $b_7$, and $b_{11}$) were applied to approximate *WQI* values, whereas other spectral indices did not have a role to play. Additionally, the total number of effective parameters and GCV value were 28.5 and 3.568, respectively. The values of WC were adjusted with particle swarm optimization (PSO) within 70 iterations and mean square error [MSE] = 7.484.

**Table 7.** Basis functions used in the development of MARS model.

| Basis Function | Formulation |
|---|---|
| $BF_1$ | $\max(0, b_1 - 276.41)$ |
| $BF_2$ | $\max(0, 0.02367 - b_1)$ |
| $BF_3$ | $\max(0, 0.1261 - b_7)$ |
| $BF_4$ | $\max(0, 0.1261 - b_7) \times \max(0, b_2 - 0.07603)$ |
| $BF_5$ | $\max(0, 0.1261 - b_7) \times \max(0, 0.07603 - b_2)$ |
| $BF_6$ | $\max(0, b_1 - 276.41) \times \max(0, b_4 - 0.08869)$ |
| $BF_7$ | $\max(0, 0.02367 - b_1) \times \max(0, b_4 - 0.0891)$ |
| $BF_8$ | $\max(0, b_1 - 276.41) \times \max(0, 287.98 - b_{11})$ |
| $BF_9$ | $\max(0, 0.02994 - b_7) \times \max(0, b_1 - 0.05777)$ |
| $BF_{10}$ | $\max(0, b_2 - 283.42)$ |
| $BF_{11}$ | $\max(0, 0.02994 - b_7) \times \max(0, b_4 - 0.08869)$ |

*4.3. Gene Expression Programming*

GEP is a powerful artificial intelligence model created from the combination and development of genetic algorithm and genetic programming by Ferreira [46]. The basic concept of GEP is the same as the genetic algorithm except that separate branches are used instead of using a single-bit strip. Each branch consists of a set of terminals and functions [47]. GEP is an evolved genotype/phenotype system where the genotype is completely separated from the phenotype. Unlike genetic programming, genotype and phenotype are combined in a frequent system. This system has a high ability to find a suitable pattern for interpreting complex systems and storing genetic data [46,47].

Solving a problem using GEP involved several steps; the first step was to select the function needed to create the model. Several statistical parameters, such as root means square error (RMSE), mean absolute error (MAE), and root relative squared error (RRSE), could be used to validate this function. In the second step, a set of functions

(i.e., mathematical operators, nonlinear functions, and terminals) were used to produce chromosomes. In the next step, an index was created to estimate the accuracy of the built model. In the fourth step, a system including numerical components and qualitative variables was determined to control the execution of the model. In the last step, the stopping criterion of the model, which could be the achievement of the desired fit or the maximum number of model executions, was included [3,48,49].

The GEP model, implemented with GeneXproTools5 software, resulted in the best relationship for predicting the water quality index. Values of genetic operators were directly dependent on the selection of the training strategies: optimal evolution, constant fine-tuning, model fine-tuning, and sub-set selection. As seen in Table 8, the best performance of GEP expression occurred for the selection of optimal evolution because this methodology benefits from the high flexibility of interaction among mathematical operators, values of genetic operators, and terminals. The Equation (17), which was obtained from four genes, was expressed as follows:

$$\text{WQI} = (b_4 + 8.6769) + \left(b_4 - \left(b_4 \times \left[\frac{1}{-3.3012 \times b_1} + b_1 - 2.09326\right]\right)\right) + \left(\frac{1 - \frac{\text{Exp}[b_7 \times 49.7054]}{2} + b_{11}}{2} + b_3\right) \quad (17)$$

**Table 8.** Setting parameters of GEP model performance.

| Parameters | Values |
| --- | --- |
| Number of chromosomes | 30 |
| Linking function | + |
| Mutation | 0.00138 |
| Fixed-Root Mutation | 0.00068 |
| Gene-Recombination | 0.00068 |
| Gene-Transportation | 0.00277 |
| One-Point Recombination | 0.00277 |
| Best fitness function | 419.5948 |
| Stop condition | R-Square Threshold |
| Maximum depth of subtree | 7 |
| Mathematical operators and function | $\pm, \times, /, \text{Ln}(x), \exp(x), \text{Average }(x_1, x_2)$ |

### 4.4. Evolutionary Polynomial Regression

EPR, as a newly-extended AI model based on regression analysis, can generally create a symbolic model to present a robust solution for the simulation of complicated behavior governing input–output systems [50]. The implementation of the EPR Multi-Objective Genetic Algorithm (MOGA) consisted of a two-stage process. First, an evolutionary algorithm was used to search for model structures. Second, a linear regression algorithm was applied to find the optimum model parameters using the least-squares technique. This multi-objective approach led to the search for optimum models while maintaining a balance between prediction accuracy and model complexity. EPR resulted in a mathematical relationship that consisted of several algebraic terms, such as [51–54]:

$$\text{WQI} = \text{bias} + \sum_{j=1}^{M} \left[\text{WC}_j \times (b_1)^{\text{ESR}(j,1)} \times \ldots \times (b_{11})^{\text{ESR}(j,11)} \times \text{H}\left((b_1)^{\text{ESR}(j,1)} \times \ldots \times (b_{11})^{\text{ESR}(j,11)}\right)\right] \quad (18)$$

in which M was the maximum number of mathematical terms, H was a user-defined-function that consisted of various mathematical structures (e.g., tangential hyperbolic, natural logarithm, and exponential functions), and ESR was a vector of exponents defined by the user.

Through the development of the EPR model, EPR expressions were provided for all typical forms of inner functions (i.e., tangent hyperbolics, natural logarithm, exponential function, and secant hyperbolic). The results of the training stages demonstrated that the use of natural logarithm was more parsimonious than other types of inner functions. On the other hand, secant hyperbolic [Sechx = $2/(e^x + e^{-x})$] and tangent hyperbolic [Tanhx = $(e^x - e^{-x})/(e^x + e^{-x})$] could provide more complicated EPR expressions in comparison with expressions given by natural logarithm and exponential functions. On many occasions, it is more suitable for engineers to select the lowest complicated expression, although its accuracy level was marginally lower than other EPR expressions. In this study, EPR expressions given by exponential function obtained the lowest accurate predictions in the training stage (MSE = 1.603) when compared with EPR expression developed with no function (MSE = 1.519), secant hyperbolic (MSE = 1.423), and tangent hyperbolic (MSE = 1.477). EPR models provided highly lower complex equations rather than equations given by secant and tangent hyperbolic functions in spite of resulting in slightly lower accurate predictions (MSE = 1.507) than hyperbolic functions. Additionally, 11 logarithm expressions were produced during the training stage of the EPR model. As seen in Table 9, each equation included six algebraic terms, and additionally, natural logarithm was employed as an inner function to approximate the WQI values due to the fact that the pollution process in the natural streams is generally a complicated process; then, applying a complex expression could improve accuracy level of predictions in comparison with employing a simple regression equation. Another important setting parameter is related to multi-objective genetic algorithm (MOGA), which is applied in the structure of the EPR model in order to optimize the number of algebraic terms, the number of variables used in the EPR model, and values of exponent dedicated to each variable. Moreover, Table 10 demonstrates the setting parameters of all EPR expressions. According to Table 9, Model.8 yielded the most accurate prediction of WQI (MSE = 1.507) in comparison with other expressions. Hence, Model.8 was elected for further analysis in the training and testing phases and robust comparisons with related works.

**Table 9.** Developed expressions by EPR models.

| Model. No | Formulation | MSE |
|---|---|---|
| 1 | $WQI = 0.0016231 \times \frac{1}{b_6^2} + 3.54 \times Ln\left(b_9^{0.5} \times b_{10}^2\right) + 5.4618 \times Ln\left(\frac{b_3^2}{b_7^2 \times b_{10}^2}\right) + 0.012942 \times b_{11} \times$ $Ln\left(\frac{1}{b_7^{1.5}}\right) + 5141.924 \times \frac{b_3^{0.5}}{b_{11}} \times Ln\left(b_7^2 \times b_{10}\right) + 63.1082 \times b_1^{0.5} \times b_7 \times Ln\left(b_6^{0.5}\right) + 128.9703$ | 1.706 |
| 2 | $WQI = 0.014943 \times \frac{1}{b_6^2} + 1.8054 \times Ln\left(b_9^{0.5} \times b_{10}^2\right) + 5.114 \times Ln\left(\frac{b_3^2}{b_7^{0.5} \times b_{10}^2}\right) + 0.010792 \times b_{11} \times$ $Ln\left(\frac{b_{10}}{b_7^{1.5}}\right) + 4814.2172 \times \frac{b_3^{0.5}}{b_{11}} \times Ln(b_7^2 \times b_{10}) + 57,955 \times b_1^{0.5} \times b_7 \times Ln\left(b_9^{0.5}\right) + 124.0877$ | 1.588 |
| 3 | $WQI = 0.0014518 \times \frac{1}{b_6^2} + 1.8395 \times Ln\left(b_9^{0.5} \times b_{10}^2\right) + 5.103 \times Ln\left(\frac{b_3^2 \times b_{11}}{b_7^{0.5} \times b_{11}^2}\right) + 0.010619 \times b_{11} \times$ $Ln\left(\frac{b_{10}}{b_7^{1.5}}\right) + 280.4126 \times \frac{b_3^{0.5}}{b_{11}^{0.5}} \times Ln\left(b_7^2 \times b_{10}\right) + 58.3367 \times b_1^{0.5} \times b_7 \times Ln\left(b_9^{0.5}\right) + 93.3195$ | 1.656 |
| 4 | $WQI = 1.8322 \times Ln\left(b_9^{0.5}\right) + 5.4262 \times Ln\left(\frac{b_3^2}{b_7^{0.5} \times b_{10}^2}\right) + 0.024901 \times b_{11} \times Ln\left(\frac{b_{10}}{b_6}\right) + 295.5584 \times \frac{b_3^{0.5}}{b_{11}^{1.5}} \times$ $Ln\left(b_7^2 \times b_{10}\right) + 280.1486 \times b_1^{0.5} \times b_7 \times Ln\left(b_1 \times b_6^{0.5}\right) + 70,121.1045 \times b_1^{0.5} \times b_6^{1.5} \times b_7 + 125.822$ | 1.585 |
| 5 | $WQI =$ $0.61063 \times Ln\left(b_9 \times b_{10}^2\right) + 5.024 \times Ln\left(\frac{b_3^2}{b_7^{0.5} \times b_{10}^2}\right) + 0.018268 \times b_{11} \times Ln\left(\frac{b_{10}}{b_6}\right) + 2077.6743 \times \frac{b_3^{0.5} \times b_6^{0.5}}{b_{11}^{0.5}} \times$ $Ln\left(b_7^2 \times b_{10}\right) + 314.7386 \times b_1^{0.5} \times b_7 \times Ln\left(b_1 \times b_6^{0.5}\right) + 76,488.31.78 \times b_1^{0.5} \times b_6^{1.5} \times b_7 + 129.5538$ | 1.585 |
| 6 | $WQI = 5.0562 \times Ln\left(\frac{b_3^2}{b_7^{0.5} \times b_{10}^2}\right) + 0.75124 \times Ln\left(b_1^{0.5} \times b_9 \times b_{10}^2\right) + 0.017321 \times b_{11} \times Ln\left(\frac{b_{10}}{b_6}\right) +$ $2103.1284 \times \frac{b_3^{0.5} \times b_6^{0.5}}{b_{11}^{0.5}} \times Ln\left(b_7^2 \times b_{10}\right) + 314.0978 \times b_1^{0.5} \times b_7 \times Ln\left(b_1 \times b_6^{0.5}\right) + 71,395.4675 \times b_1^{0.5} \times$ $b_6^{1.5} \times b_7 + 133.1729$ | 1.521 |

**Table 9.** *Cont.*

| Model. No | Formulation | MSE |
|:---:|:---:|:---:|
| 7 | $\text{WQI} = 4.852 \times \text{Ln}\left(\frac{b_3^2}{b_7^{0.5} \times b_{10}^2}\right) + 0.78368 \times \text{Ln}\left(b_1^{0.5} \times b_2^{0.5} \times b_9 \times b_{10}^2\right) + 0.016237 \times b_{11} \times \text{Ln}\left(\frac{b_{10}}{b_6}\right) + 2022.4638 \times \frac{b_3^{0.5} \times b_6^{0.5}}{b_{11}^{0.5}} \times \text{Ln}\left(b_7^2 \times b_{10}\right) + 306.1721 \times b_1^{0.5} \times b_7 \times \text{Ln}\left(b_1 \times b_6^{0.5}\right) + 66{,}319.6416 \times b_1^{0.5} \times b_6^{1.5} \times b_7 + 133.5218$ | 1.58 |
| 8 | $\text{WQI} = 4.9538 \times \text{Ln}\left(\frac{b_3^2}{b_7^{0.5} \times b_{10}^2}\right) + 0.017425 \times b_{11} \times \text{Ln}\left(\frac{b_{10}}{b_6}\right) + 4.8373 \times b_6^{0.5} \times \text{Ln}\left(b_1^{0.5} \times b_2^{0.5} \times b_9 \times b_{10}\right) + 2061.653 \times \frac{b_3^{0.5} \times b_6^{0.5}}{b_{11}^{0.5}} \times \text{Ln}\left(b_7^2 \times b_{10}\right) + 303.5518 \times b_1^{0.5} \times b_7 \times \text{Ln}\left(b_1 \times b_6^{0.5}\right) + 61{,}065.493 \times b_1^{0.5} \times b_6^{1.5} + b_7 + 132.2393$ | 1.499 |
| 9 | $\text{WQI} = 4.6907 \times \text{Ln}\left(\frac{b_3^2}{b_7^{0.5} \times b_{10}^2}\right) + 0.015951 \times b_{11} \times \text{Ln}\left(\frac{b_{10}}{b_6}\right) + 5.7518 \times b_7^{0.5} \times \text{Ln}\left(b_1^{0.5} \times b_2^{0.5} \times b_3^{0.5} \times b_9 \times b_{10}^2\right)0.5 + 34{,}537.1755 \times \frac{b_3^{0.5} \times b_6^{0.5}}{b_{11}} \times \text{Ln}\left(b_7^2 \times b_{10}\right) + 336.6602 \times b_1^{0.5} \times b_7 \times \text{Ln}\left(b_1 \times b_6^{0.5}\right) + 75{,}604.5548 \times b_1^{0.5} \times b_6^{1.5} \times b_7 + 133.6116$ | 1.602 |
| 10 | $\text{WQI} = 4.7174 \times \text{Ln}\left(\frac{b_3^2}{b_7^{0.5} \times b_{10}^2}\right) + 0.01511 \times b_{11} \times \text{Ln}\left(\frac{b_{10}}{b_6}\right) + 6.1317 \times b_7^{0.5} \times \text{Ln}\left(b_1^{0.5} \times b_2^{0.5} \times b_3^{0.5} \times b_9 \times b_{10}^2\right) + 36{,}315.0134 \times \frac{b_3^{0.5} \times b_6^{0.5}}{b_{11}} \times \text{Ln}\left(b_7^2 \times b_{10}\right) + 334.1929 \times b_1^{0.5} \times b_7 \times \text{Ln}\left(b_1 \times b_6^{0.5}\right) + 299{,}003.26 \times b_1^{0.5} \times b_3^{0.5} \times b_6^{1.5} \times b_7 + 136.9156$ | 1.562 |
| 11 | $\text{WQI} = 4.6505 \times \text{Ln}\left(\frac{b_3^2 \times b_{11}^{0.5}}{b_7^{0.5} \times b_{10}^2}\right) + 0.014827 \times b_{11} \times \text{Ln}\left(\frac{b_{10}}{b_6}\right) + 6.1112 \times b_7^{0.5} \times \text{Ln}\left(b_1^{0.5} \times b_2^{0.5} \times b_3^{0.5} \times b_9 \times b_{10}^2\right) + 35{,}910.17 \times \frac{b_3^{0.5} \times b_6^{0.5}}{b_{11}} \times \text{Ln}\left(b_7^2 \times b_{10}\right) + 335.2054 \times b_1^{0.5} \times b_7 \times \text{Ln}\left(b_1 \times b_6^{0.5}\right) + 303{,}430.10.54 \times b_1^{0.5} \times b_3^{0.5} \times b_6^{1.5} \times b_7 \times 123.4055$ | 1.507 |

**Table 10.** Setting parameters of EPR model performance.

| Inner Function | Natural Logarithm |
|:---:|:---:|
| Range of exponents | $[-2, -1.5, -1, -0.5, 0, 0.5, 1, 1.5, 2]$ |
| Number of terms | 6 |
| Expression structure | $\text{Sum}(a_i \times x_1 \times x_2 \times f(x_1 \times x_2)) + \text{bias}$ |
| Regression method | Non-negative least squares |
| Optimum number of Generation | $[10\ 40]$ |
| Fitness function | Mean Square Error |

## 5. Results and Discussion

### 5.1. Statistical Performance of Soft Computing Techniques

The results of the quantitative evaluation of the training and testing phases are shown in Table 11. As seen in Table 11, the MARS model indicated the highest level of accuracy (IOA = 0.992 and RMSE = 0.0.640) in the prediction of WQI for the training phase when compared with GEP (IOA = 0.964 and RMSE = 1.383), MT (IOA = 0.969 and RMSE = 1.287), and EPR (IOA = 0.973 and RMSE = 1.194) models. Additionally, values of MAE (0.0059) and SI (0.0073) proved the superiority of the MARS model [Equation (16)] over other AI models: GEP (MAE = 0.0104 and SI = 0.0157), MT (MAE = 0.0091 and SI = 0.00146), and EPR (MAE = 0.0076 and SI = 0.0135). Figure 3a illustrates the qualitative performance of AI models in the training stage. According to Figure 3a, for observed values of WQI = 82–87, all the predicted values were concentrated on the best-fit line. Additionally, the majority of the data points were in the acceptable range of WQI error predictions ($\pm25\%$). EPR and MT models indicated over-prediction of WQI values for observed values of WQI = 90–93. It can be inferred that these AI models have the weakest performance compared to MARS and GEP models.

**Table 11.** Performance of various AI models in the training and testing phase for prediction of WQI.

| AI Models | Training Phase | | | |
|---|---|---|---|---|
| | IOA | RMSE | MAE | SI |
| MT | 0.969 | 1.287 | 0.0091 | 0.0146 |
| MARS | 0.992 | 0.64 | 0.0059 | 0.0073 |
| GEP | 0.964 | 1.383 | 0.0104 | 0.0157 |
| EPR | 0.973 | 1.194 | 0.0076 | 0.0135 |
| AI Models | Testing Phase | | | |
| | IOA | RMSE | MAE | SI |
| MT | 0.978 | 1.085 | 0.0084 | 0.0146 |
| MARS | 0.975 | 1.165 | 0.0088 | 0.0129 |
| GEP | 0.978 | 1.052 | 0.0093 | 0.0109 |
| EPR | 0.977 | 1.123 | 0.0083 | 0.0135 |

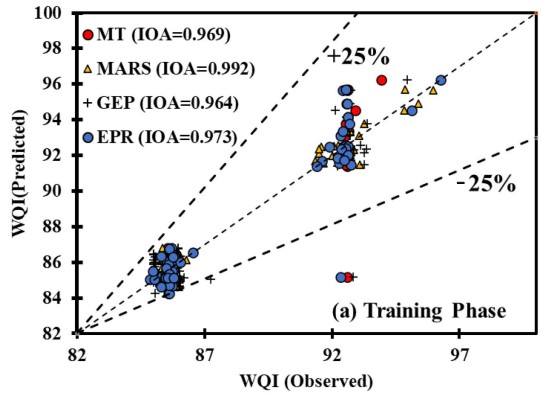 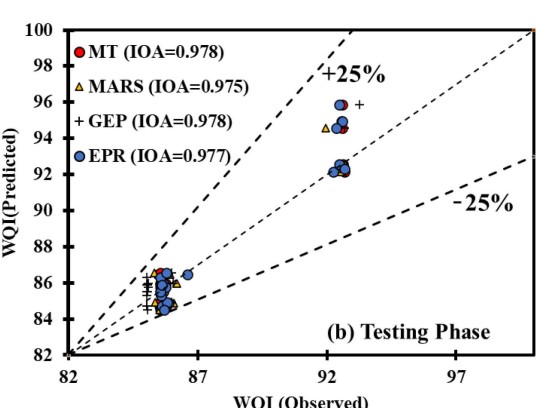

**Figure 3.** Performance of AI models in the prediction of WQI for (**a**) training and (**b**) testing phases.

The results of the testing phase were given in Table 11. According to statistical criteria, Equation (17), given by the GEP model, provided a rather more accurate prediction of WQI (IOA = 0.980 and RMSE = 1.053) than MT (IOA = 0.978 and RMSE = 1.085), EPR (IOA = 0.977 and RMSE = 1.123), and MARS (IOA = 0.975 and RMSE = 1.165). Although IOA values were close together, MAE values had marginal differences. Moreover, the SI value given by the GEP model (SI = 0.0109) was slightly lower than MARS (SI = 0.0129), EPR (SI = 0.0135), and MT (SI = 0.0146). This means a rather higher performance of the GEP model in the testing phase than other AI models. Equations (13) and (14), given by MT, provided a slightly more precise estimation of WQI values (IOA = 0.978, RMSE = 1.085, and MAE = 0.0084) in comparison with the MARS model (IOA = 0.975, RMSE = 1.165, and MAE = 0.0088). In fact, multivariate linear regression equations given by MT are quite simple rather than the second-order polynomial expression [Equation (16)] by the MARS model. Furthermore, GEP and EPR models provided more accurate predictions with complicated mathematical structures (e.g., natural logarithm and exponential functions) in comparison with MT. The qualitative performance of AI models in the testing phase has been depicted in Figure 3b. Although AI models demonstrated the overprediction of WQI values for the observed WQI values between 90 and 93, all the predicted values of WQI ranged in the permissible error band.

In order to comparatively express the efficacy of the present predictive tools, the analysis of the violin plots was employed. Generally, both training and testing datasets

were utilized to draw violin plots. Relative Error (RE) values for all AI models have been computed to evaluate the distribution of error values along with AI models performance:

$$\text{RE} = \frac{1}{U} \sum_{i=1}^{U} \frac{\text{WQI (i)}_{\text{Obs}} - \text{WQI(i)}_{\text{Pre}}}{\text{WQI(i)}_{\text{Obs}}} \tag{19}$$

From Figure 4, it was found that all violin plots were relatively symmetrical. RE values given by the MARS model had a rather narrower range (0.35–0.90) than MT (0.4–1.1), EPR (0.2–0.95), and GEP (0.4–1.3) models. In addition to this, the median of RE values given by the MARS plot is lower when compared to other violin plots. As seen in Figure 4, violin plots presented by the MARS model demonstrated that a large number of RE values intend to perfect value (zero) in comparison with other EPR, MT, and GEP models. Moreover, the distribution of RE values given by EPR and MT models was relatively identical. The maximum width of violin plots produced by the EPR and MARS models were relatively the same at RE = 0.4, whereas the maximum ones obtained 0.8 and 0.75 for the GEP and MT, respectively.

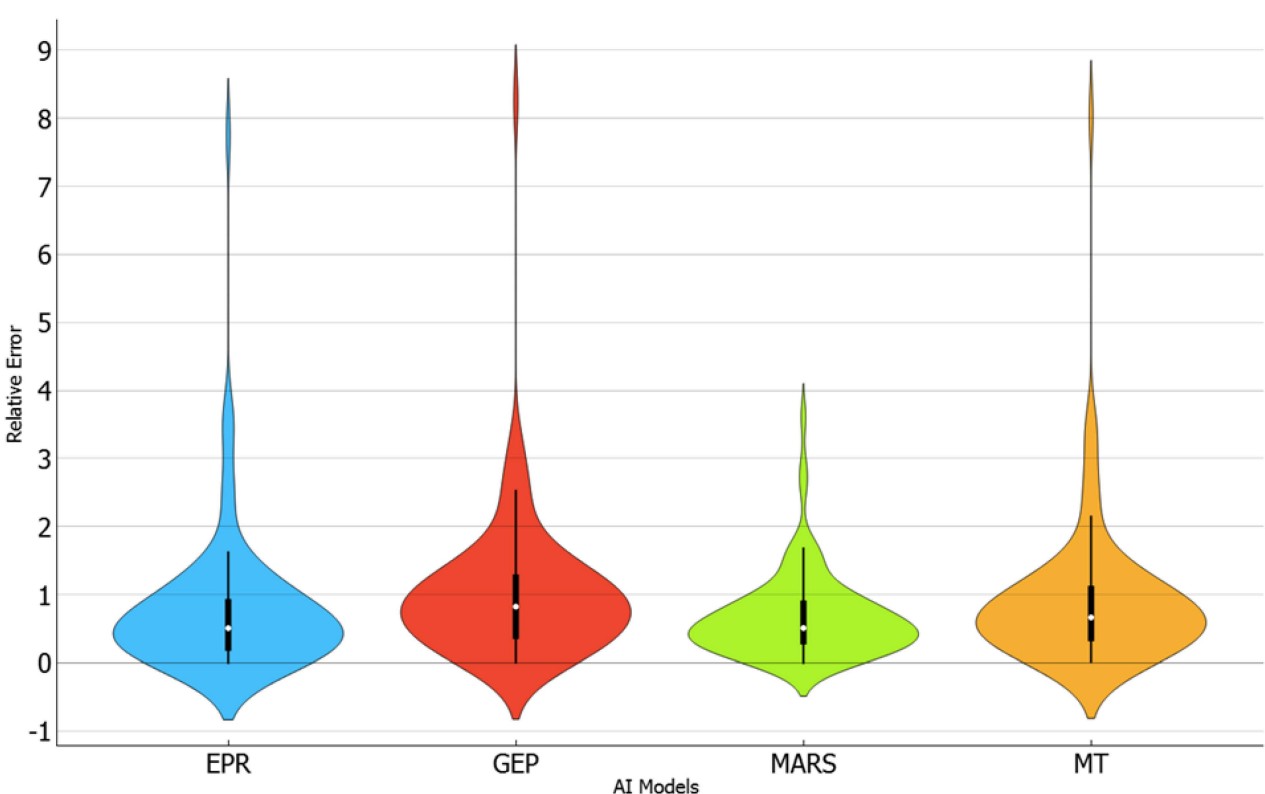

**Figure 4.** Comparison of the performance of AI models with violin plot.

The usability of the statistical measures (i.e., IOA, RMSE, MAE, and SI) is likely to fail to fully understand the efficacy of AI models in order to approximate WQI values. In this way, this study employs the Fisher test (*F*-test) to deeply investigate the evaluation of AI models' performance. The chief aim of the *F*-test is to define whether the hypothesis claiming that "the value of variation calculated with respect to the regression model is greater compared to that value computed based on averages" is acceptable. In order to obtain this major, *F*-test utilizes the *F*-ratio ($F_0$). Accordingly, the null hypothesis of the *F*-test stands at the acceptable level when $F_0 > F_{\alpha, \gamma, \lambda}$ in which $\alpha$ is the significant level (0.05) and $\lambda$ is the number of spectral indices ($\gamma = 11$), and $\lambda$ denotes $U - \gamma - 1$ (95−11−1 = 83). In addition to this, $F_0$ is calculated by $\text{MS}_R / \text{MS}_E$ where $\text{MS}_R$ [SSR/($\gamma$−1)] and $\text{MS}_E$ [SSE/(U−$\gamma$−1)] denote the mean square regression and the mean square error, respectively. SSR and SSE

denote the sum of squares regression and the sum of squares error respectively that are computed as follows:

$$\text{SSR} = \sum_{i=1}^{U} (WQI(i)_{Pre} - WQI(i)_{Obs})^2 \tag{20}$$

$$\text{SSE} = \sum_{i=1}^{U} (WQI(i)_{Pre} - \overline{WQI}_{Obs})^2 \tag{21}$$

In this way, $F_{0.05,12,83}$ is roughly equal to 2.112. Table 12 indicated the results of *F*-test for all AI models. As inferred from Table 13, MARS ($F_0 = 0.327$), MT ($F_0 = 1.513$), and GEP ($F_0 = 0.8771$) accept the hypothesis of the *F*-test, whereas the EPR model did not satisfy the hypothesis ($F_0 = 5.639$).

**Table 12.** Results of *F*-test for AI models.

| AI Models | SSR | SSE | MSR | MSE | $F_0$ | Hypothesis States |
|-----------|-----|-----|-----|-----|-------|-------------------|
| GEP | 170.861 | 1213.8 | 13.143 | 14.985 | 0.877 | Accept |
| MARS | 60.549 | 1150.9 | 4.657 | 14.208 | 0.327 | Accept |
| EPR | 12462 | 13770 | 958.587 | 169.995 | 5.639 | Reject |
| M5MT | 308.959 | 1272.20 | 237.766 | 15.706 | 1.513 | Accept |

**Table 13.** Results of uncertainty analysis for AI models.

| AI Models | $\mu_e$ | $S_e$ | $CL_e^+$ | $CL_e^-$ | Uncertainty Band $(CL_e^+ - CL_e^-)$ |
|-----------|---------|-------|----------|----------|--------------------------------------|
| GEP | 0.0710 | 0.8152 | 0.1419 | 0.0000003 | 0.1419 |
| MARS | 0.1732 | 1.5702 | 0.2755 | 0.0710 | 0.2046 |
| EPR | 0.2252 | 1.9852 | 0.2771 | 0.1732 | 0.1039 |
| M5MT | 0.2997 | 2.3734 | 0.3741 | 0.2252 | 0.1489 |

*5.2. Complexity of AI Model-Derived Expressions*

The complexity of AI model-based formulations is directly dependent on the number of setting parameters and the methodologies tuning the parameters. In the GEP model, there is a wide range of genetic operators that play a key role in controlling the accuracy level of the GEP formulation. These operators (i.e., mathematical and genetic operators) would provide more complex expressions rather than mathematical expressions given by the MARS and EPR models. On the other hand, the setting parameters of the EPR model were selected before running the model, and it will cause them to reduce the complexity of the expression extracted by EPR. In contrast, the mathematical expression of the GEP model changes continuously during the GEP performance because a wide range of setting parameters was employed. In GEP, determining the genetic operations have four strategies: optimal evolution, constant fine-tuning, model fine-tuning, and sub-set selection. Among these operators, the selection of optimal evolution is more suitable for finding function problems than other operators, although the complexity of the GEP expression increases. Additionally, the MT has the fastest performance compared to the other AI models. In MT, the first set of MT-based equations was diminished after smoothing and tuning the tree of MT. Then, Equations (13) and (14) were generated for the simplest form in comparison with expressions given by GEP and EPR models. According to the EPR applications in the WQPs/WQI predictions, it was confirmed that the application of EPR expression with natural logarithmic inner function could better detect the complexity between WQI and WQPs rather than the EPR regression model without an inner function. Moreover, applying three inner functions (exponential, secant, and tangent hyperbolic functions) provided more complex EPR expressions, although the results of the training stages were

comparatively accurate. Furthermore, the EPR model included three general mathematical structures in order to define interactions among input variables: $y_1 = Sum[ai \cdot x_1 \cdot x_2 \cdot f(x_1 \cdot x_2)]$, $y_2 = Sum[ai \cdot f(x_1 \cdot x_2)]$, and $y_3 = Sum[ai \cdot x_1 \cdot x_2 \cdot f(x_1) \cdot f(x_2)]$. This study employed y1 to receive more accurate predictions of WQI, although the usability of y1 increased the complexity of EPR expressions when compared with y2 and y3.

### 5.3. Variation of WQI Values by AI Models

Figure 5 illustrates spatial changes in the values of the predicted WQI by AI models for the image taken on 12 March 2021. From Figure 5a–d, it is clear that the WQI values vary between 81.84 and 88.87, indicating a good grade of surface water quality. In addition, the spatial variations given by AI models are relatively the same. As seen in Figure 5, WQI values gradually decreased from the northern section of the reach to the near vicinity of the river reach middle section, then; the WQI values indicate an upward trend. From west to east, the water quality increases for all AI models. Moreover, Figure 6 depicts only temporal variations of the predicted WQI by MT for all dates from 12 March 2021 (dd/mm/yy) to 7 June 2021 because temporal variations given by AI models have a relatively similar trend. As seen in Figure 6a,b, WQI values gradually increased from 12 March 2021 to 13 April 2021 and then slightly plummeted on 20 April 2021 (Figure 6c). From Figure 6c&d, the class of surface water quality has remained relatively constant (Class II). After that, the class of WQI values increased to stand at class I for the excellent state on 15 May 2021, as illustrated in Figure 6e. In the last month, Figure 6f indicated WQI had a slight decrease compared to that predicted in the previous month (15 May 2021). Overall, WQI values given by MT vary between 90 and 93.93.

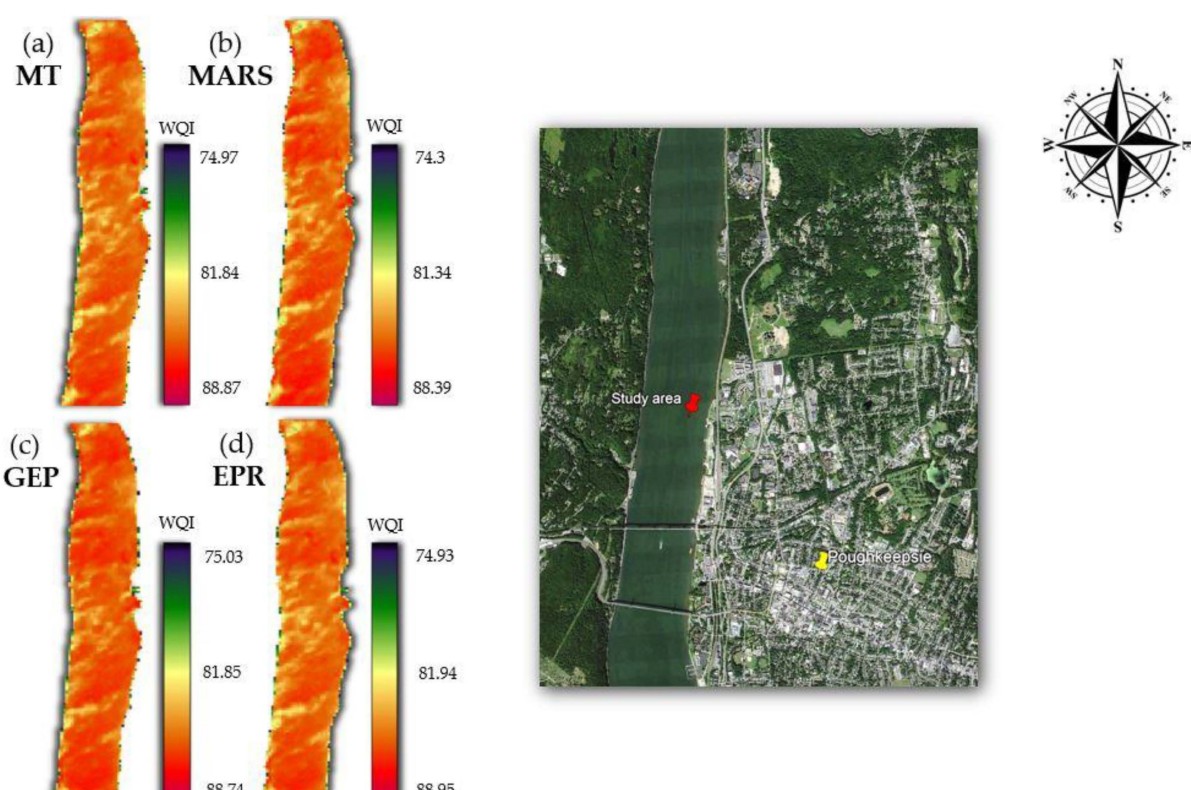

**Figure 5.** Spatial variations of WQI predicted by AI models for 12/03/2021: (**a**) MT, (**b**) MARS, (**c**) GEP, and (**d**) EPR.

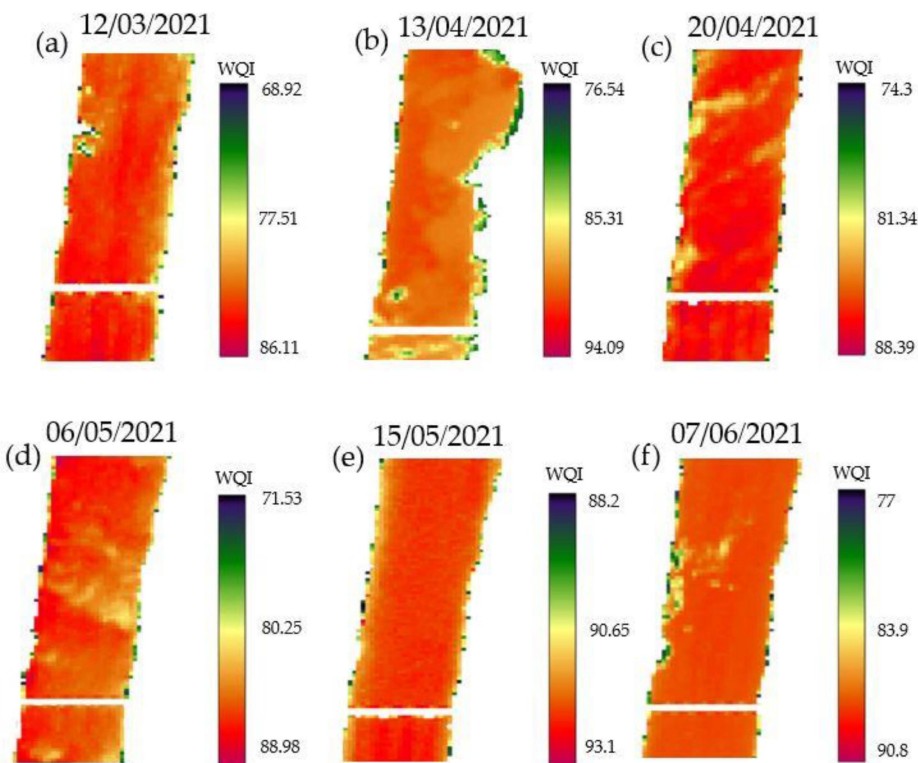

**Figure 6.** Temporal variations of WQI predicted by MT for all dates of satellite images: (**a**) 12/03/2021, (**b**) 13/04/2021, (**c**) 20/04/2021, (**d**) 06/05/2021, (**e**) 15/05/2021, and (**f**) 07/06/2021.

Additionally, the receiver-operating characteristic (ROC) curves are used to evaluate the overall performance of AI models for both the training and testing stages. In order to derive the ROC curve, the meaning of sensitivity and specificity should be understood. These concepts are directly applied to evaluate the performance of AI models. After that, the area under the curve (AUC) needs to be computed. Detailed descriptions of ROC curves were presented in the literature [55]. For this purpose, the WQI values predicted by the AI models were served as the model's predictions, whereas the turbidity observation data were employed as the AI model's control values. In this study, the observed water quality index was chosen as a control value. Figure 7 illustrated the ROC curves for all AI models. From Figure 7, it is clear that all AI models have excellent performance with an AUC greater than 0.97.

In order to quantify the uncertainty related to the AI models (i.e., MT, MARS, GEP, and EPR), the confidence bands of estimation errors ($CL_e^{\pm}$) are computed as follows [56,57]:

$$CL_e^{\pm} = \mu_e \pm Z_a . S_e \tag{22}$$

in which $\mu_e$ is the mean of estimation errors, and $S_e$ is the standard deviation of estimation errors; $Z_a$ is the standard normal variable at the 5% of significant level. In order to make comparisons among the uncertainty values given by the AI models in this study, the $CL_e^{\pm}$ values at the 5% of the significant level for all datasets (i.e., training and testing datasets) have been provided in Table 13. From Table 13, it is clear that the AI models result in overestimated predictions ($\mu_e > 0$) for WQI values: GEP (0.0710), EPR(0.2252), MARS(0.1732), and M5MT (0.2997). Additionally, the lowest value of estimation uncertainty is given by the EPR model with an uncertainty band of 0.0710, whereas the M5MT generates the highest level of uncertainty (0.2997). Generally, the findings of Table 13 demonstrate that EPR expression has the most superior performance when compared with other AI models applied in the current study.

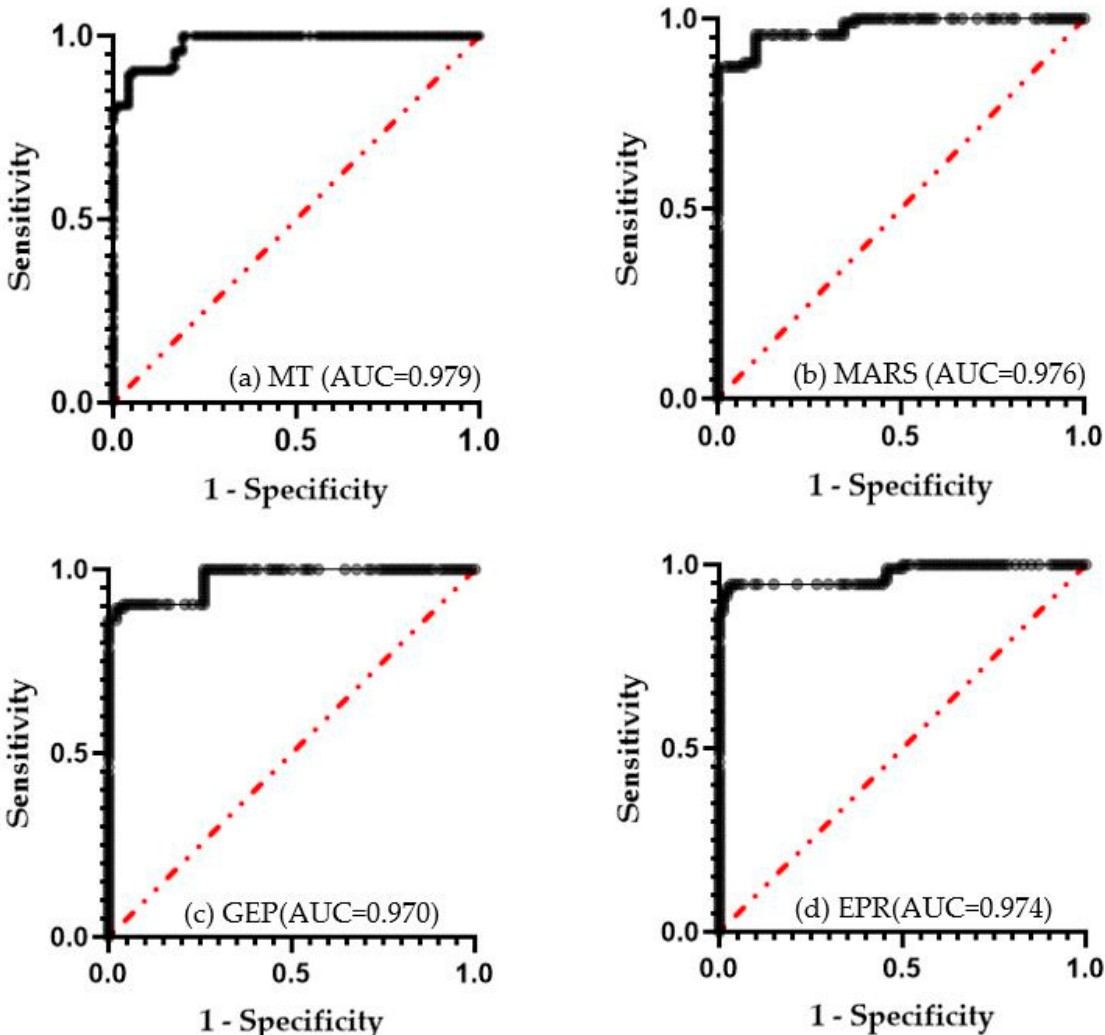

**Figure 7.** ROC curve of AI models for predicting WQI: (**a**) MT, (**b**) MARS, (**c**) GEP, and (**d**) EPR.

*5.4. Comparisons of the Present Study with the Literature*

In this section, the results of the present study were compared with the relevant literature in terms of various facts: complexity of AI models, accuracy levels of AI models, and restrictions of satellite images.

Chebud et al. [22] applied ANN and seven bands of Landsat-8 spectral data as input parameters. They proposed ANN with five hidden neurons in order to predict three WQPs (i.e., Chlorophyll-a, turbidity, and phosphorus). The ANN given by Chebud et al. [22] was introduced as a black-box model, whereas the present AI models in this study were white-box with a high interpretability of information. On the contrary, the present study reported WQI as a good indicator of various WQPs compared with those WQPs investigated in Chebud et al. [22]. Additionally, MT [Equations (13) and (14)] and MARS [Equation (16)] predicted WQI value lower complicated mathematical expressions rather than the ANN model (7-5-3) by Chebud et al.'s [22] investigations. Zhang et al. [21] employed three spectral indices of Sentinel-2 images (DI, RI, and NDI) as input parameters to predict WQI values with the SVM model. From their investigations, the useability of the spectral indices of water increased the complexity of the SVM model because these indices were computed with combinations of b values along with various derivative orders (0–3). In contrast, the present study did not apply combinations of b values due to the decrease in the computational volume of AI models. In terms of accuracy levels, SVM given by Zhang et al. [21] predicted the WQI values with R = 0.9 and RMSE = 213.41, called

the best results with a derivative order of 3. To make a rigid judgment, the present AI models provided WQI values with a high degree of precision [MARS (R = 0.965 and RMSE = 1.165), GEP (R = 0.969 and RMSE = 1.052), EPR (R = 0.969 and RMSE = 1.123), and MT (R = 0.969 and 1.085]] in comparison with Zhang et al. [21] results. Moreover, linear and second-order regression equations provided expressions in comparison with SVM given by Zhang et al. [21].

Ariad-Rodriguez et al. [26] utilized powerful machine learning models (SVM and ELM) and linear regression equations in order to monitor the turbidity of surface water resources. They used satellite images given by Landsat-8 OLI. From their study, SVM with the sigmoid kernel (R = 0.748 and RMSE = 27.75) and ELM (R = 0.3464 and RMSE = 9.16) estimated turbidity with lower accuracy when compared with the present results. ELM was structured by 10,000 neurons in the hidden layer, providing a highly complex network for turbidity predictions. On the other hand, linear regression by the least square method obtained R = 0.8402 and RMSE = 97.64 by considering $b_2$, $b_5$, $b_6$, and $b_7$ as input parameters. The linear regression equation by Arias-Rodriguez et al. [26] had lower performance than the multivariate linear equation by MT (R = 0.969 and RMSE = 10.85). The present study applied AI models (i.e., GEP, EPR, MARS, and MT) to provide an accurate and less complicated model compared with ELM and SVM models by Arias-Rodriguez et al. [26], who proved that a remote sensing-based GP model had the acceptable capability to predict TP with R = 0.761. They used MODIS satellite images, which were not capable of retrieving WQPs with accurate predictions and finer spatial resolution, as well as Landsat-8.

In Chang et al.'s [23] investigations, the precision level of the GP model was lower than the present results, such as GEP and EPR with R = 0.969. As a merit, GEP models used a multi-genes system to provide a non-linear regression equation, which demonstrated a better performance than the GEP model, although mathematics expressions given by GEP and EPR models were complicated. Chen et al. [27] applied six AI models [CatBoost (R = 0.9246 and RMSE = 3.120), XGBoost (R = 0.9143 and RMSE = 4.231), AdaBoost (R = 0.9192 and RMSE = 4.822), RF (R = 0.912 and RMSE = 5.031), DNN (R = 0.7823 RMSE = 6.347), and KNN (R = 0.8837 and RMSE = 5.730)] to provide turbidity by predictions, as well as AI models in this study. The AI models given in Chen et al.'s [27] study come from black-box and complicated structures when compared with AI models in this study.

Li et al. [19] applied five machine learning models (SVM, RF, ANN, Regression Three [RT], Gradient Boost Machine [GBM]) to predict the total nitrogen (TN) and total phosphate (TP) using Landsat-8 images. From their study, the performance of five AI models indicated low accuracy level for TN [SVM (R = 0.449), RT (R = 0.7), ANN (R = 0.6708), RT (R = 0.4123), and GBM (R = 0.5)] and TP [SVM (R = 0.7681), RT (R = 0.4582), ANN (R = 0.8185), RT (R = 0.4898), and GBM (R = 0.6480)] predictions when compared with the present study. Furthermore, Li et al. [19] used spectral indices of satellite images (e.g., RI, DI, and NDI) to establish correlations between WQPs and b values. This issue plummeted the accuracy level of AI models in the prediction of WQPs in comparison with the present study.

## 6. Conclusions

In this study, AI models were created to estimate 13 WQPs using Landsat-8 images and observational data of the Hudson River. A dataset containing the estimated values of the quality characteristics was built by utilizing the developed models and applying them to satellite images. The creation of the newly produced data set was used to obtain the CCME water quality index. Four artificial intelligence techniques, MT, MARS, GEP, and EPR, were utilized to construct a relationship to estimate the water quality index. Overall, the main findings of the current study were drawn as:

- The correlation coefficients of WQP with single bands revealed that a considerable number of parameters were highly correlated with Landsat-8 bands 10 and 11;
- The correlation between spectral data and WQP improves when spectral indexes (RI and NDI) are utilized. In addition, the results showed that the use of spectral indices in some cases led to an increase in the value of R2 in MLR models;

- The WQI values were computed from the observed water quality data, which varied from 84.2 to 96.25 in the Hudson River. The observed WQI values given by CCME guidelines were indicative of good state of quality;
- The WQI values were predicted with AI models, for which four robust expressions were provided based on eight bands of Landsat-8 images. All the AI models were developed along with the optimum selection of the setting parameters;
- Statistical measures (i.e., IOA, RMSE, MAE, and SI) quantified the satisfying performance of non-linear multivariate expressions given by AI models (i.e., EPR, GEP, and MARS) and linear regression model (MT) in the prediction of WQI values for both training and testing stages. In addition, the results of the F-test and AUC approved the quantitative performance, and more importantly, the qualitative efficiency of AI models was statistically studied with violin graphs. Moreover, the uncertainty results of AI models performance indicated that EPR and MT had the lowest and highest degrees of uncertainty;
- AI models could efficiently detect both spatial and temporal variations of the WQI values for the studied reach of the Hudson River. Additionally, the comparisons of the present results with the literature were done in terms of the accuracy levels of AI models, the structural complexity of AI models, and the typical use of satellite images. According to R and RMSE criteria, the results of the present AI models (i.e., EPR, MT, GEP, and MARS) as white-box models were comparable with studies performed with SVM, RF, ANN, RT, and GBM models (introduced as black-box models).

In this study, we investigated a practical and economical way of assessing the quality status of the river. For less developed nations that are dealing with issues like inadequate equipment, poor budgets, etc., the combination of satellite imagery with AI to assess water quality may be a particularly acceptable solution.

**Supplementary Materials:** The following are available online at https://www.mdpi.com/article/10.3390/rs15092359/s1, Table S1: Values of correlation coefficients between spectral bands and WQPs; Table S2: Correlation coefficients between ratios of band indices and WQPs.

**Author Contributions:** Conceptualization, M.N. and S.B.; Methodology, M.N. and S.B.; Software, M.N. and S.B.; Validation, M.N.; Formal Analysis, M.N. and S.B.; Investigation, M.N.; Writing—Original Draft Preparation, M.N. and S.B.; Writing—Review and Editing, M.N.; Visualization, M.N. and S.B.; Supervision, M.N. All authors have read and agreed to the published version of the manuscript.

**Funding:** This research received no external funding.

**Institutional Review Board Statement:** Not applicable.

**Informed Consent Statement:** Not applicable.

**Data Availability Statement:** Not applicable.

**Acknowledgments:** This research has been supported by the Graduate University of Advanced Technology (Kerman-Iran) under grant number of 1650.

**Conflicts of Interest:** The authors declare no conflict of interest.

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
