# Peer review of "Evaluation of River Water Quality Index Using Remote Sensing and Artificial Intelligence Models"

_remotesensing, doi:10.3390/rs15092359_

Round 1

Reviewer 1 Report (Previous Reviewer 2)

The authors have correctly adressed my comments and the paper can be accepted.

Reviewer 2 Report (Previous Reviewer 1)

The authors appropriately considered my suggestions.

This manuscript is a resubmission of an earlier submission. The following is a list of the peer review reports and author responses from that submission.

Round 1

Reviewer 1 Report

The study is generally well-written and I only suggest the following moderate revision/changes.

1- As stated in the WQI estimation, there are two global approaches to approximate WQI, called CCME and NSF guidelines. Why did authors use CCME rather than NSF?

2- In the MARS modeling, Which curve-fitting approach was used to approximate coefficients of the polynomial regression splines? It seems authors applied least square techniques, if so this issue should be clarified.

3- Expression given by GEP model can become more complex than those obtained by EPR (polynomial regression via multi-objective genetic algorithm) and MARS (polynomial regression spline through second order analysis). This should be explained.

4- There are 11 expressions by EPR model which have been obtained on the basis of natural logarithms. Why did not scholars employ "no function" option through the development of EPR?

5- Why did the authors apply statistical measures (IOA, RMSE, MAE, and SI) to evaluate efficiency of AI models: EPR, MARS, MT, and GEP? All these should be justified.

6- I recommend involving some of the following studies which can be useful for the journal readers.

https://doi.org/10.3390/w14223636

https://doi.org/10.1016/j.compag.2018.04.019

https://doi.org/10.1007/s11069-017-3123-9

https://doi.org/10.1016/j.jhydrol.2017.03.020

Reviewer 2 Report

While the paper is within the scope of the journal, I can clearly see that the scientific contribution is very limited. Modelling water quality index using machine learning is well reported and discussed in the literature and several high quality paper are now published. The paper was relatively poorly strutured and in overall, the authors have not provided the necessary efforts for preparing a high quality paper, which can help in improving our understanding in this subject. My comments about the paper are as follow:

1.      The authors are invited for improving the writing style of the paper and this can be achieved by deeply checking the literature review.

2.      It is clear that the WQI was modelled using several water quality variables which is extremely obvious. It is clear that an ensemble of water quality variables were highly correlated to the WQI, thus any ML model should provide good predictive accuracies.

3.      Models evaluation and comparison without the R, R2, and NSE, is unacceptable at this level of publication.

4.      Machine learning models are not well presented.

5.      The paper is without discussion section which extremely unacceptable

6.      Section results is very poor and out of scope.

7.      Results should be presented with more figures, which is not the case.
